# Identification and Functional Evaluation of Polyphenols That Induce Regulatory T Cells

**DOI:** 10.3390/nu14142862

**Published:** 2022-07-13

**Authors:** Tsukasa Fujiki, Ryosuke Shinozaki, Miyako Udono, Yoshinori Katakura

**Affiliations:** 1Faculty of Pharmaceutical Sciences, Nagasaki International University, 2825-7 Huis Ten Bosch, Sasebo, Nagasaki 859-3298, Japan; fujiki@niu.ac.jp; 2Graduate School of Bioresources and Bioenvironmental Sciences, Kyushu University, 744 Motooka, Nishi-ku, Fukuoka 819-0395, Japan; shinozaki@s.kyushu-u.ac.jp; 3Faculty of Agriculture, Kyushu University, 744 Motooka, Nishi-ku, Fukuoka 819-0395, Japan; mudono@grt.kyushu-u.ac.jp

**Keywords:** regulatory T cells, retinaldehyde dehydrogenase, IgA, quercetin, luteolin

## Abstract

Regulatory T cells (Tregs) and CD4^+^/CD25^+^ T cells play an important role in the suppression of excessive immune responses, homeostasis of immune function, and oral tolerance. In this study, we screened for food-derived polyphenols that induce Tregs in response to retinaldehyde dehydrogenase (*RALDH2*) activation using macrophage-like THP-1 cells. THP-1 cells were transfected with an EGFP reporter vector whose expression is regulated under the control of mouse *Raldh2* promoter and named THP-1 (Raldh2p-EGFP) cells. The THP-1 (Raldh2p-EGFP) cells were treated with 33 polyphenols after inducing their differentiation into macrophage-like cells using phorbol 12-myristate 13-acetate. Of the 33 polyphenols, five (kaempferol, quercetin, morin, luteolin and fisetin) activated *Raldh2* promoter activity, and both quercetin and luteolin activated the endogenous *Raldh2* mRNA expression and enzymatic activity. Furthermore, these two polyphenols increased transforming growth factor beta 1 and forkhead box P3 mRNA expression, suggesting that they have Treg-inducing ability. Finally, we verified that these polyphenols could induce Tregs in vivo and consequently induce IgA production. Oral administration of quercetin and luteolin increased IgA production in feces of mice. Therefore, quercetin and luteolin can induce Tregs via *RALDH2* activation and consequently increase IgA production, suggesting that they can enhance intestinal barrier function.

## 1. Introduction

Regulatory T cells (Tregs) and CD4^+^/CD25^+^ T cells play an important role in suppressing excessive immune responses, maintaining homeostasis of immune function [1], and further regulating oral immune tolerance [2]. There are two types of Tregs: naturally occurring Tregs, which are directly differentiated from undifferentiated cells in the thymus, and induced Tregs (iTregs), which are differentiated from naive CD4^+^ T cells upon antigen stimulation in peripheral tissues, such as the intestinal tract. iTregs are considered important in the regulation of antigen-specific immune responses in the periphery. Retinoic acid (RA) has been found to be involved in the induction of Treg and Th17 cell differentiation and in the regulation of immune cell differentiation and function [3]. Additionally, RA promotes the differentiation of forkhead box P3 FOXP3^+^ iTregs and inhibits the differentiation of Th17 cells in a transforming growth factor beta (TGF)-β-dependent manner [3].

After Vitamin A is converted from retinyl ester into retinol in the liver, it is released into the bloodstream, where it binds to retinol-binding proteins and circulates in the body. Retinaldehyde dehydrogenase (RALDH) catalyzes the conversion of retinol into RA. RALDH1 to RALDH3 exist as isoforms of RALDH; dendritic cells in intestine-related tissues mainly express *RALDH2* [4]. Intestinal dendritic cells and mucosal intrinsic layer macrophages produce RA in a RALDH2-dependent manner, and activation of the *RALDH2* gene in these cells plays an important role in Treg induction [5,6]. At present, no studies have evaluated Treg induction via *RALDH2* activation mediated by food components. Therefore, we aimed to identify polyphenols that activate *RALDH2* expression and further evaluate their function in vivo. We found that quercetin and luteolin can induce Tregs via *RALDH2* activation and consequently increase IgA production, suggesting that they can enhance intestinal barrier function.

## 2. Materials and Methods

### 2.1. Cell Culture and Reagents

THP-1 cells of human acute monocytic leukemia were cultured in RPMI 1640 medium (Nissui Pharmaceutical, Tokyo, Japan) supplemented with 10% fetal bovine serum (FBS; Life Technologies, Gaithersburg, MD, USA) at 37 °C and 5% CO_2_. All polyphenols were purchased from Fujifilm Wako Pure Chemical (Osaka, Japan). All polyphenols were dissolved in dimethyl sulfoxide (DMSO) at the concentration of 10 mM. These polyphenol stocks were diluted 1000-fold and added to the cells. DMSO was used as a control.

### 2.2. Establishment of a Reporter System to Screen for Polyphenols That Activate the Raldh2 Promoter

Polymerase chain reaction (PCR) was performed using the primers 5′-ATTAATAACTGACTTACCAGCTCGT-3′ and 5′-GCTAGCGGCGATCTCGCTGGAAGTCA-3′ and mouse genomic DNA to clone the mouse *Raldh2* promoter. After replacing the CMV promoter of the EGFP-C3 vector with the *Raldh2* promoter, the vector was transfected into THP-1 cells. Transfected cells were selected using 800 μg/mL of G418 (Fujifilm Wako Pure Chemical, Osaka, Japan) to establish a stable cell line of THP-1 (Raldh2p-EGFP cells).

### 2.3. Screening of Polyphenols That Activate Raldh2 Promoter via IN Cell Analyzer 2200

Effects of the polyphenols on *Raldh2* promoter activity in differentiated THP-1 cells were evaluated by monitoring changes in enhanced green fluorescent protein (EGFP) fluorescence derived from THP-1 (Raldh2p–EGFP) cells using the IN Cell Analyzer 2200 (Cytiva, Tokyo, Japan). THP-1 (Raldh2p–EGFP) cells were seeded in 96-well blackplates (Greiner Bio-one, Tokyo, Japan) at a density of 6 × 10^5^ cells/mL, treated with 100 ng/mL phorbol 12-myristate 13-acetate (PMA), and cultured for 48 h. After culturing, polyphenols were directly added to the cells at the final concentration of 10 μM and cells were further cultured for 24 h. Cells were then fixed with 4% formaldehyde for 15 min at room temperature. After washing the cells with phosphate-buffered saline (PBS), the cells were stained with 1 μg/mL Hoechst 33,342 solution (Dojindo, Kumamoto, Japan) for 20 min. The relative EGFP fluorescence intensity per cell was measured using IN Cell Analyzer 2200.

### 2.4. Quantitative Reverse Transcription-PCR (RT-qPCR)

THP-1 cells were seeded in 60 mm dishes at a density of 6 × 10^5^ cells/mL, induced to differentiate via addition of 100 ng/mL PMA, and then subsequently cultured for 48 h at 37 °C. Cells were then cultured in the presence of 10 μM of polyphenol for 48 h. RNA was isolated using High Pure RNA Isolation kit (Roche Diagnostics GmbH, Mannheim, Germany), and cDNA was prepared using ReverTra Ace qPCR RT Master Mix (Toyobo, Osaka, Japan) according to the manufacturer’s instructions. RT-qPCR was performed using Thunderbird SYBR qPCR mix (Toyobo) and Thermal Cycler Dice Real Time System TP-800 (TaKaRa Bio, Shiga, Japan). The samples were analyzed in triplicate, and gene expression levels were normalized to the corresponding β-actin levels. The PCR primer sequences used were as follows: human β-actin: forward primer 5′-TGGCACCCAGCACAATGAA-3′ and reverse primer 5′-CTAAGTCATAGTCCGCCTAGAAGC-3′; *Raldh2*, forward primer 5′-GCAATGCAAGCTGGGACTGT-3′ and reverse primer 5′-CCCGCAAGCCAAATTCTCCC-3′; *TGFB1*, forward primer 5′-AACCGGCCTTTCCTGCTTCT-3′ and reverse primer 5′-ACGCAGCAGTTCTTCTCCGT-3′; *FOXP3*, forward primer 5′-AGTGGCCCGGATGTGAGAAG-3′ and reverse primer 5′-ACATTGTGCCCTGCCCTTCT-3′.

### 2.5. Flow Cytometry

THP-1 cells were differentiated using 100 ng/mL PMA and then treated with 10 μM quercetin or 10 μM luteolin for 24 h. The ALDEFLUOR reagent system (Stemcell Technologies, Cambridge, MA, USA) was used to monitor cellular aldehyde dehydrogenase activity using a CytoFlex flow cytometer (Beckman Coulter, Miami, FL, USA).

### 2.6. Preparation of Human Peripheral Blood Mononuclear Cells (PBMCs)

PBMCs were isolated from collected human peripheral blood using Leucosep (Greiner Bio-one). Cells were washed with PBS, seeded into 5 mL dishes at a cell density of 1.0 × 10^6^ cells/mL, and cultured in RPMI 1640 medium containing 10% FBS for 24 h. On the next day, cells were seeded into 24-well plates at a cell density of 1.0 × 10^6^ cells/mL and cultured in the presence of 10 μM polyphenol for 24 h. RNA preparation, cDNA synthesis and qRT-PCR were performed according to the methods described in Section 2.4.

### 2.7. Animal Experiments

Seven-week-old male BALB/c mice (Japan SLC Co., Ltd., Shizuoka, Japan) were assigned to six groups (*n* = 6) and orally administered with luteolin and quercetin at 0.2 and 2 mg/kg body weight, respectively. Mice were fed food and water ad libitum, and oral administration was performed once a day at 10:00 AM. Mice were housed individually for 7 d in a 12 h:12 h light/dark cycle at 23 °C and 60% humidity. Feces were collected daily before oral administration. The collected feces were weighed, suspended in PBS containing protease inhibitor cocktail, and centrifuged, and the supernatant was collected and stored at −85 °C. All animal experiments were conducted in accordance with the “Guidelines for the Handling and Use of Animals” of Nagasaki International University.

### 2.8. Measurement of Fecal IgA Content via Enzyme-Linked Immunosorbent Assay (ELISA)

The amount of IgA secreted into the intestinal tract of mice was measured via ELISA. Mouse fecal samples were dissolved in PBS using cOmplete Mini protease inhibitor cocktail (Roche Diagnostics GmbH). A microtiter plate (Nunc, Naperville, IL, USA) was coated with anti-mouse IgA antibody (eBioscience, Burlingame, CA, USA) diluted in 0.1 M sodium carbonate buffer (pH 9.6) and incubated at 37 °C for 2 h. The plate was washed three times with PBS containing 0.05% Tween 20 (PBS-T). The supernatant of the mouse fecal solution was serially diluted and added to the plate, which was incubated overnight at 4 °C. After washing three times with PBS-T, diluted horseradish peroxidase-conjugated goat anti-mouse IgA antibody (eBioscience) was added and the plate was incubated for 2 h at 37 °C. After washing five times with PBS-T, the 3,3′,5,5′-tetramethylbenzidine substrate (eBioscience) was added and the plate was incubated at room temperature for 15 min. The absorbance at 450 nm was measured using an ELISA plate reader.

## 3. Results

### 3.1. Screening for Polyphenols That Activate the Raldh2 Promoter

We screened for polyphenols that activate the *Raldh2* promoter in differentiated THP-1 (Raldh2p-GFP) cells. Changes in fluorescence intensity derived from THP-1 (Raldh2p-GFP) cells after adding polyphenols were monitored. Among the 33 polyphenols tested, five polyphenols including kaempferol, quercetin, morin, luteolin and fisetin were found to activate the *Raldh2* promoter (Figure 1A).

Next, the effects of these polyphenols on the expression of endogenous *RALDH2* in differentiated THP-1 cells were evaluated via RT-qPCR. Among the five polyphenols, quercetin and luteolin significantly increased the expression of endogenous *RALDH2* mRNA in THP-1 cells (Figure 1B).

### 3.2. Effects of the Raldh2-Affecting Polyphenols on THP-1 Cells

Next, we tested the effects of *RALDH2*-affecting polyphenols on THP-1 cells. First, changes in RALDH2 enzyme activity upon polyphenol treatment were evaluated using the ALDEFLUOR reagent system which can detect enzymatic activity of aldehyde dehydrogenase including RALDH2. Differentiated THP-1 cells were treated with 10 μM quercetin or luteolin and incubated for 24 h. We quantified the change in RALDH2 activity upon quercetin and luteolin treatment using the change in median fluorescence as an indicator of RALDH2 activity. The results showed that treatment with quercetin and luteolin increased the median fluorescence intensity by 25.8% and 15.4%, respectively, compared to the control, indicating that the respective polyphenols enhanced RALDH2 activity (Figure 2A,B).

Next, we determined the effects of the two polyphenols on the expression of TGF-β in THP-1 cells. TGF represents an important cytokine involved in Treg induction. Quercetin and luteolin treatment significantly affected *TGFB1* mRNA levels (Figure 2B). These results suggest that quercetin and luteolin may activate macrophages by affecting *RALDH2* expression and inducing TGF expression, thereby creating an environment in which Tregs can be further induced.

### 3.3. Effect of RALDH2 Promoter-Activating Polyphenols on Treg Induction

We investigated whether macrophage-activating polyphenols can induce Tregs using human PBMCs. Potential Treg induction in PBMCs via polyphenol treatment was determined based on FOXP3 expression, a known marker of Treg cells, as an indicator. PBMCs were incubated with quercetin or luteolin for 24 h, and endogenous *FOXP3* expression level was determined via qRT-PCR. Quercetin and luteolin treatment significantly induced endogenous *FOXP3* in PBMCs (Figure 3), indicating that these polyphenols could induce Tregs in vivo.

### 3.4. Effect of Orally Administered Polyphenols on Treg Function In Vivo

Several studies have shown that Tregs contribute to IgA production in vivo. Therefore, we tested whether polyphenols, which could activate macrophages and induce Tregs in PBMCs, can enhance *Raldh2* expression and induce IgA production as a result of Treg induction in vivo. We tested the effect of polyphenols on IgA production by measuring the amount of IgA in the feces of mice orally administered with polyphenols. Compared to the control, quercetin and luteolin treatment significantly induced IgA production in vivo. In particular, quercetin treatment strongly induced IgA production even at low concentrations (Figure 4). Therefore, quercetin and luteolin can induce *Raldh2* expression in macrophages and induce Tregs, consequently inducing IgA production in vivo. However, as shown in Figure 4, dose-dependent results could not be obtained with quercetin in particular. This may be due to differences in the in vivo environment, differences in local concentration-dependent availability, or polymerization, etc. We aim to verify this point in future studies.

## 4. Discussion

Previous studies have shown that RA induces Treg differentiation and inhibits Th17 differentiation [1,3]. Therefore, this study focused on RALDH, which is known to be involved in RA synthesis, to identify polyphenols that activate the *RALDH2* gene and to clarify its function. THP-1 cells induced to differentiate into macrophage-like cells via PMA treatment were used as the cell line for tracking changes in *RALDH2* expression.

Screening revealed that five polyphenols (kaempferol, quercetin, morin, luteolin and fisetin) increased *Raldh2* promoter expression, two of which (quercetin and luteolin) enhanced endogenous *RALDH2* expression in differentiated THP-1 cells. The bioactivities of quercetin and luteolin have been reported, including inhibition of cholesterol absorption in the intestinal tract, strengthening of the intestinal barrier mediated by quercetin [7,8], and the antidepressant effect of luteolin [9]. Furthermore, these polyphenols have been reported to increase the number of Tregs, increase the production of Treg-related cytokines, and reduce arthritis via anti-inflammatory effects in a mouse model of rheumatoid arthritis [10]. In vitro, luteolin has been reported to exhibit anti-inflammatory effects in mouse models of enteritis and dextran sulfate sodium-induced colitis [11].

The function of quercetin and luteolin was evaluated in this study. RA is known to play an important role in IgA production by inducing B cell homing to the intrinsic layer of the small intestine and expression of α4β7 and C-C chemokine receptor 9 [12]. The two polyphenols identified in this study also enhanced *RALDH2* expression and induced Tregs, suggesting that oral administration of quercetin and luteolin to mice may increase IgA production in the intestine and suppress inflammation [13]. The IgA content in the feces of mice treated with these two polyphenols was increased, indicating that the polyphenols induced Tregs and enhanced IgA production in the intestinal tract as a result of *RALDH2* induction. Although other polyphenols, such as isoliquiritigenin and naringenin, exhibit Treg-inducing activity [1], we demonstrated that quercetin and luteolin induce Tregs and consequently induce intestinal IgA production as well as *RALDH2* enhancement in this study. These two polyphenols are thought to be responsible for the enhancement of barrier function and defense against infections via the enhancement of IgA production in the intestinal tract. The detailed molecular mechanisms of the enhancement of *RALDH2* expression by quercetin and luteolin should be clarified in the future. Furthermore, any additional functions of these polyphenols mediated via Treg induction should be elucidated.

## Figures and Tables

**Figure 1 nutrients-14-02862-f001:**
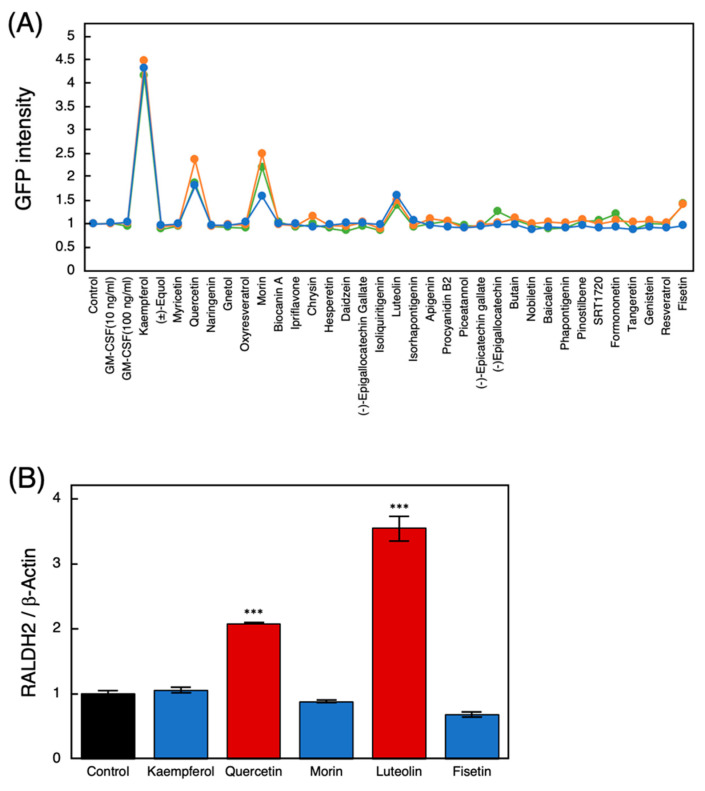
Screening for polyphenols that activate the *Raldh2* promoter in THP-1 cells. (**A**) THP-1 (Raldh2p-GFP) cells differentiated using phorbol 12-myristate 13-acetate were treated with 33 polyphenols (10 μM) and incubated for 24 h. Changes in EGFP fluorescence were monitored using IN Cell Analyzer 2200 (green, orange and blue show three independent experiments). (**B**) The effect of quercetin and luteolin on the expression of endogenous *RALDH2* mRNA in differentiated THP-1 cells was assessed via quantitative reverse transcription-polymerase chain reaction. Dimethyl sulfoxide (DMSO) was used as a control. Two-sided Student’s *t*-test was used to test for significant differences compared to the results of the controls. Significance was defined as *** *p* < 0.001. Raldh2: retinaldehyde dehydrogenase 2; EGFP: enhanced green fluorescent protein.

**Figure 2 nutrients-14-02862-f002:**
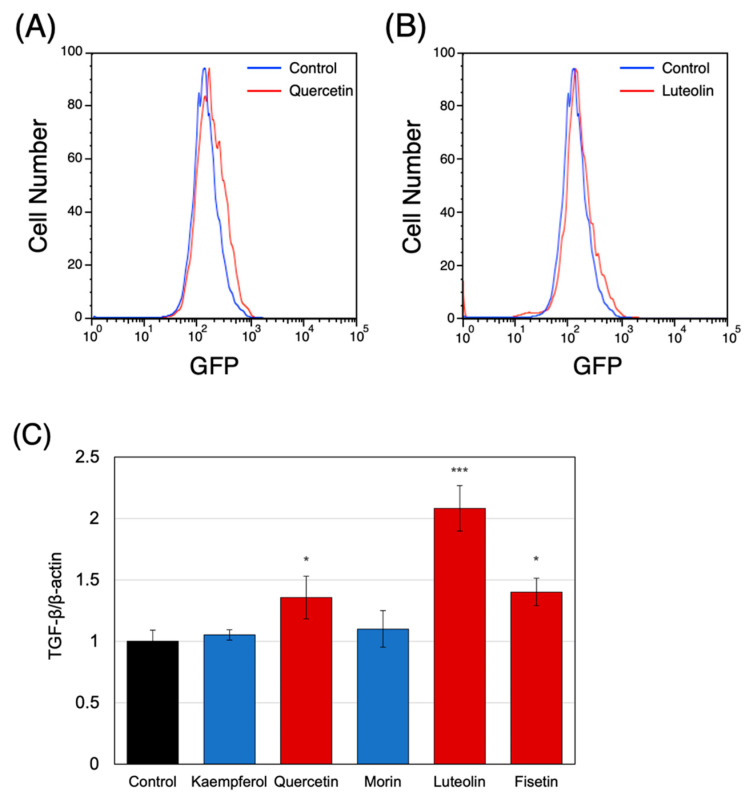
Quercetin and luteolin activated the endogenous RALDH enzymatic activity in THP-1 cells. Enzymatic activity of RALDH2 in differentiated THP-1 cells treated with (**A**) quercetin and (**B**) luteolin were measured using the ALDEFLUOR staining kit. (**C**) Quercetin and luteolin treatment increased the *TGFB1* mRNA expression level in the differentiated THP-1 cells. Two-sided Student’s *t*-test was used to test for significant differences compared to results of the controls. Significance was defined as * *p* < 0.05, *** *p* < 0.001. RALDH2: retinaldehyde dehydrogenase 2; TGFB1: transforming growth factor beta 1.

**Figure 3 nutrients-14-02862-f003:**
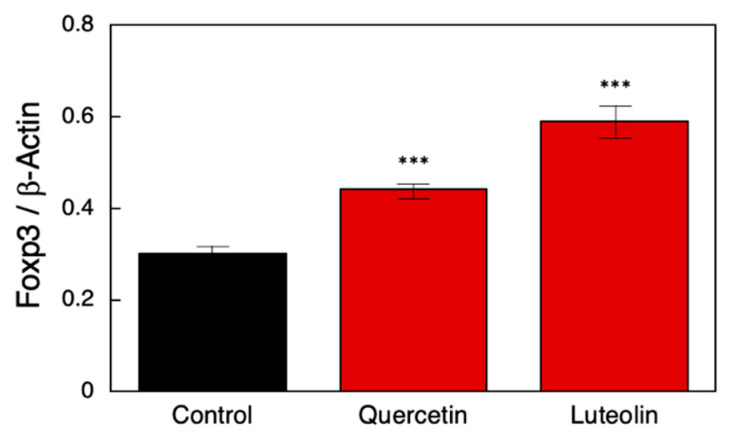
Quercetin and luteolin treatment affected *FOXP3* mRNA expression in PBMCs. *FOXP3* mRNA levels were determined via quantitative polymerase chain reaction in triplicate. Two-sided Student’s *t*-test was used to test for significant differences compared to results of the controls. Significance was defined as *** *p* < 0.001. FOXP3: forkhead box P3; PBMCs: peripheral blood mononuclear cells.

**Figure 4 nutrients-14-02862-f004:**
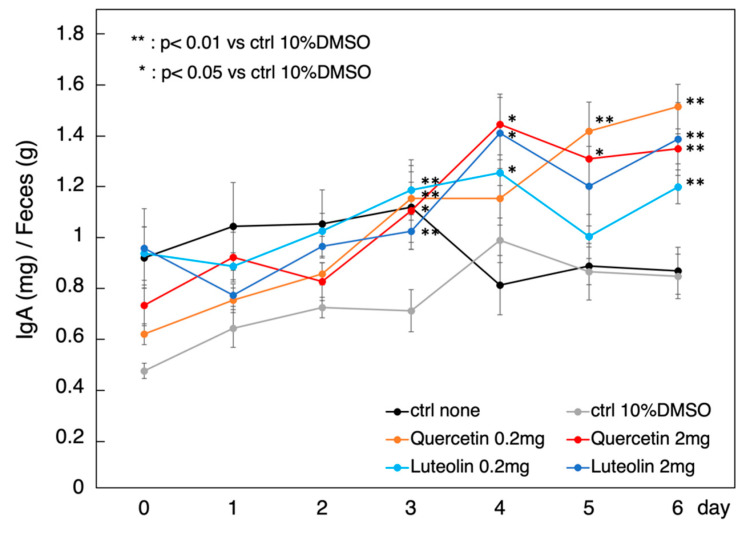
Effect of oral administration of *Raldh2*-activating polyphenols on IgA production in the intestines of BALB/c mice. Quercetin and luteolin were administered orally to BALB/c mice (*n* = 6), and total IgA was measured via enzyme-linked immunosorbent assays using mouse fecal samples. IgA expression is expressed as the mean ± standard error of mean mg/g fecal weight using a two-sided Student’s *t*-test. Significant differences were tested in comparison to results of the controls. Significance was defined as * *p* < 0.05, ** *p* < 0.01. *R**aldh2*: retinaldehyde dehydrogenase.

## Data Availability

The data that support the findings of this study are available from the corresponding author, Y.K., upon reasonable request.

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
