# Peer review of "Identification and Functional Evaluation of Polyphenols That Induce Regulatory T Cells"

_nutrients, 2022, doi:10.3390/nu14142862_

Round 1

Reviewer 1 Report

This current study conducted by the group of Yoshinori Katakura has shown that two polyphenols named quercitin and luteolin can induce regulatory T cells through the activation of retinaldehyde 13 dehydrogenase (RALDH2) which consequently increased the generation of IgA. The authors have identified these two phenols amongst the screened 33 polyphenols after the primary screening in Vitro. It’s a well-written manuscript and easily understandable. However, I suggest to clarify the following issues

1.       In the experimental section, 2.3, did you wash the cells (or changed the culture medium) before treating the cells with the polyphenols? Also, what was the control? Was it just only the medium?

2.       Could you please include the method of preparation of the polyphenols?

3.       In the result section, 3.2, Fig 2A and 2B, could you please clarify the difference between control and the quercetin and luteolin?

4.       In figure 2C, the authors have included the activity of fisetin. I suggest the authors include the flow cytometry data for the fisetin as well? Also, could you please explain why you discarded the further study of fisetin even though it shows similar activity compared to quercitin in Fig 2C?

Author Response

Response to the comments of Reviewer 1

Thank you for your valuable comments. I revised the manuscript according to the reviewer’s comments.

Comment #1:

In the experimental section, 2.3, did you wash the cells (or changed the culture medium) before treating the cells with the polyphenols? Also, what was the control?

Response to the comment #1:

Thank you for your comments. Before answering to your question, in Section 2.3, we have corrected the incubation time from 48 h to 24 h. We apologize for this correction.

The polyphenols were directly added to the cells without washing the cells.

DMSO (dimethyl sulfoxide) was used as a control.

Accordingly, we revised the manuscript.

Comment #2:

Could you please include the method of preparation of the polyphenols?

Response to the comment #2:

Thank you for your comment. All 33 polyphenols were dissolved in DMSO at the concentration of 10 mM. These polyphenol stocks were diluted 1000-fold and added to the cells.

Accordingly, we revised the manuscript.

Comment #3:

In the result section, 3.2, Fig 2A and 2B, could you please clarify the difference between control and the quercetin and luteolin?

Response to the comment #3:

Thank you for your comment.

We quantified the change in RALDH2 activity upon quercetin and luteolin treatment using the change in median fluorescence as an indicator of RALDH2 activity.

Accordingly, we revised the manuscript.

Comment #4:

In figure 2C, the authors have included the activity of fisetin. I suggest the authors include the flow cytometry data for the fisetin as well? Also, could you please explain why you discarded the further study of fisetin even though it shows similar activity compared to quercitin in Fig 2C?

Response to the comment #4:

Thank you for your comments.

In Fig. 1, we found that quercetin and luteolin activated RALDH2 promoter and increased RALDH2 mRNA expression in THP-1 cells and also activated endogenous RALDH2 enzyme activity.

However, although Fisetin enhanced TGF-β mRNA expression, it did not increase RALDH2 promoter activity and mRNA expression. Therefore, we do not believe that fisetin increases RALDH2 activity and have not tested the effect of fisetin on RALDH2 activity.

Tregs are induced in a RALDH2- or TGF-β-dependent manner (Guo, A.;et. Al., 2015, Iwata M;2009), and quercetin and luteolin are thought to induce Tregs through activation of both. Fisetin, on the other hand, only potentiates TGF-β, so we judged that its activity to induce Tregs is low.

Reviewer 2 Report

It is interested that polyphenols(quercetin and luteolin) induce regulatory T cells with RALDH2.

There are some minor concerns authors need to pay attention:

I would like to see the de novo synthesis in vitro and dose dependent change in vivo.

Especially the IgA production of quercetin is not dose dependent.

Author Response

Response to the comments of Reviewer 2

Thank you for your valuable comments. I revised the manuscript according to the reviewer’s comments.

 Comment #1:

I would like to see the de novo synthesis in vitro and dose dependent change in vivo.

Especially the IgA production of quercetin is not dose dependent.

Response to the comment #1:

Thank you for your comment. We intend to study its effect on IgA de novo synthesis in vitro, based on the establishment of a validation system for this purpose, in the future study.

Fig. 4 shows the data of increased IgA production by oral administration of quercetin and luteolin to BALB/c mice, but as you pointed out, dose dependent results could not be obtained with quercetin in particular.

This may be due to differences in the in vivo environment, differences in local concentration-dependent availability, or polymerization, etc. We intend to verify this point in the future.

Accordingly, we revised the manuscript.